# On the Origin of the Blue Color in The Iodine/Iodide/Starch Supramolecular Complex

**DOI:** 10.3390/molecules27248974

**Published:** 2022-12-16

**Authors:** Szilárd Pesek, Maria Lehene, Adrian M. V. Brânzanic, Radu Silaghi-Dumitrescu

**Affiliations:** Department of Chemistry, Faculty of Chemistry and Chemical Engineering, Babeş-Bolyai University, 11 Arany Janos Street, 400028 Cluj-Napoca, Romania

**Keywords:** amylose, iodine, iodide, starch, UV-vis, DFT, TD-DFT

## Abstract

The nature of the blue color in the iodine-starch reaction is still a matter of debate. Some textbooks still invoke charge-transfer bands within a chain of neutral I_2_ molecules inside the hydrophobic channel defined by the interior of the amylose helical structure. However, the consensus is that the interior of the helix is not altogether hydrophobic—and that a mixture of I_2_ molecules and iodide anions reside there and are responsible for the intense charge-transfer bands that yield the blue color of the “iodine-starch complex”. Indeed, iodide is a prerequisite of the reaction. However, some debate still exists regarding the nature of the iodine-iodine units inside the amylose helix. Species such as I_3_^-^, I_5_^-^, I_7_^-^ etc. have been invoked. Here, we report UV-vis titration data and computational simulations using density functional theory (DFT) for the iodine/iodide chains as well as semiempirical (AM1, PM3) calculations of the amylose-iodine/iodide complexes, that (1) confirm that iodide is a pre-requisite for blue color formation in the iodine-starch system, (2) propose the nature of the complex to involve alternating sets of I_2_ and I_x_^-^ units, and (3) identify the nature of the charge-transfer bands as involving transfer from the I_x_^-^ σ* orbitals (HOMO) to I_2_ σ* LUMO orbitals. The best candidate for the “blue complex”, based on DFT geometry optimizations and TD-DFT spectral simulations, is an I_2_-I_5_-I_2_ unit, which is expected to occur in a repetitive manner inside the amylose helix.

## 1. Introduction

The nature of the blue complex of amylose with iodine (λ_max_ = 615 nm) has been a focus of research for many decades [1]. The left-handed helix, with an outer diameter of ~13 Å and a pitch of 8 Å (six 1,4-glucose units per pitch), hosts an internal cavity of ~5 Å where the iodine alongside iodide is hosted with I–I distances of ~3.1 Å [2,3,4,5,6,7,8,9,10]. Slight differences in the blue color are seen depending on the size (and, implicitly, biological source) of amylose; similar complexes are seen with many related poly and oligo saccharides, or even with other organic polymers [11,12,13,14,15,16,17,18,19,20,21].

The stoichiometry and charge of the poly-iodine substructures of the starch-iodine complex is still disputed. The initial supposition was that the cavity of the starch helix serves as a nonpolar solvent [22], where I_2_ molecules then dissolve in the hydrophobic interior as discrete units connected by non-covalent interactions, and that these non-covalent interactions afford the intense blue color. Semiempirical intermediate neglect of differential overlap configuration interaction (INDO CI) simulations of UV-vis spectra have been used as an argument to support an entirely neutral character poly-I_2_ nature of the blue complex (more specifically, a (C_6_H_10_O_5_)_16.5_ · I_6_ formula), and to refute I_n_^-^ (*n* = 3, 5, or 7) as possible species in the blue amylose-iodine complex [23]. However, it is now generally accepted that iodide ions are also required in the process, so that the poly-iodine chains inside the amylose helix consist of a mixture/combination of I_2_ and I^-^ [1,2,3,4,5,24] with an unusual metallic-like structure [25]. The manner in which contiguous chains of such polyanionic structures can be reconciled with a hydrophobic nature of the interior cavity of the helix, especially without including counterions, at any point throughout the cavity, may be argued to remain an interesting puzzle. Indeed, based on thermodynamic considerations it has been predicted that tri-iodide units would be unable to form long chains inside the amylose helix unless cooperativity with I_2_ molecules occurs; furthermore, the ends of the helix would in this scenario remain unoccupied by iodine atoms [26]. Also importantly, the secondary structure of amylose does vary depending on the environment (e.g., solvent, ionic strength, pH, temperature, surfactants, affecting among others the internal diameter of the helix)—to the extent that alcohol-precipitated amylose in the solid state can efficiently bind molecular I_2_ vapors (and, unlike in aqueous solution, no iodide is needed) [18,21,27,28,29]. Interestingly, however, this amylose-I_2_ complex, unlike the iodine-iodide one, is not stable when dissolved in water [30], though the preparation of an amylose-iodine complex in water at higher temperatures in the absence of iodide has also been reported [31]. Inclusion complexes of I_2_ with other oligosaccharides in the absence of iodide have also been described [32].

Various lengths of the poly-iodine chain have been proposed based on experimental data, from 3–4 to 14–15 to as high as 160 [1,2,3,4,5]. Based on potentiometric titration at low iodide concentrations, a 3/2 I_2_/I^-^ ratio was proposed, leading to an I_8_^2-^ empirical formula. However, many other chain lengths have been proposed, such as I_4_^-^, I_7_^-^, I_9_^-^, I_6_^2-^, I_8_^2-^, I_10_^2-^, I_4_^2-^, I_6_^-^, and I_24_^2-^—especially as the structure (as observable e.g., in UV-vis and circular dichroism spectra) appears to be distinctly dependent on the iodide and iodine concentrations [3,19,20,33,34,35,36]. Moreover, based on stopped-flow UV-vis and circular dichroism (CD) kinetic data, a dynamic nature of complex formation was shown, with shorter chains of iodine entering the amylose helix very fast (less than 1 millisecond), and then rearranging rapidly without further interaction with iodine/iodide from the exterior of the helix [2,37]. Interestingly, at conditions where the UV-vis spectrum of the blue complex is stable in time, its optical rotatory dispersion (ORD) signal is seen to still increase, suggesting more complex dynamics of the helix that do not affect the length of the poly-iodine chains [38]. Based on the structure of (benzamide)_z_ H^+^I_3_^-^, a poly-I_3_ structure has been proposed, and a range of kinetic, spectroscopic (UV-vis, CD, Raman, X-ray absorption) and thermodynamic data have been interpreted within this framework [15,17,39,40]. However, X-ray diffractometric data has suggested that the exclusive occurrence of either of the single species I_2_ or I_3_^-^ is unlikely [41,42,43]. Based on similarities between the Raman and Mössbauer spectra of the starch–iodine complex and those of polycrystalline (trimesic acid · H_2_O)_10_H^+^I_5_^-^, an I_5_^-^ structure was proposed for the amylose-iodine complex [44,45]. In the Raman spectra of starch–iodine complexes, four principal peaks at 27, 55, 109, and 160 cm^−1^ were observed. Using theoretical calculations and far-infrared data, the peak at 109 cm^−1^ was assigned to I_3_^-^ regarded as an impurity while I_5_^-^ would be the dominant species [46,47,48,49,50]. A role for water molecules in controlling the structure of the iodine-amylose complex was also noted [51], as was the fact that that that iodine binding does to a certain degree affect the structure of amylose–both geometrically (in terms of the dimensions of the helix, as well as in terms of the helical vs. random coil secondary structure within the amylose polymer) and in terms of rigidity [13,52,53,54,55].

Reported here is an experimental and computational study of the amylose-iodine-iodide system, using UV-vis spectra, time-dependent density functional theory (TD-DFT), and semiempirical and molecular mechanics calculations in order to model the structure of the complex and to assess the viability of various candidates potentially responsible for the blue color. Poly-I_2_ as well as I_n_^-^ structures are examined. Poly-I_2_ structures are found to afford charge-transfer bands that amplify the intensity of the color compared with an isolated I_2_ molecule, but this is still significantly below the intensity observed experimentally for the amylose-iodine-iodide mixture. On the other hand, I_2_-I_n_^-^ mixtures afford significantly stronger charge-transfer bands, from the I_x_^-^ σ* orbitals (HOMO) to I_2_ σ* (LUMO) orbitals. These bands are most likely responsible for the well-known intense blue color of the amylose-iodine-iodide system.

## 2. Results and Discussion

### 2.1. Amylose-Iodine/Iodide Complexes: Structural Considerations

As detailed in the Introduction and confirmed by experiments shown in Appendix A, I_2_ alone, either in aqueous or in alcoholic solutions (methanol or ethanol) leads to barely detectable changes in absorbance at ~600 nm in reaction with starch. Iodine-iodide mixtures do yield a blue color, which is not observed with glucose instead of starch. The absorbance maximum of the iodine-iodide-starch complex in the experiments reported in Appendix A is at ~580 nm. This is slightly different from the 615 nm reported for amylose but is in line with the fact that our starch samples also contain amylopectin, as well as with the fact that the precise position of the maximum depends on the source of the starch as well as on the relative concentrations of the reagents [3,17,35,36,39,56]. Indeed, slight concentration-dependent changes in the position of the maximum are observed, as seen in Appendix A, when varying the iodide concentration as well as when varying the length of the amylose/amylopectin chains by employing an amylase.

Figure 1 and Appendix A show the results of geometry optimizations on helical amylose models where the internal cavity is either empty (model A), or filled with water or with iodine (I_2_, I_3_^-^ or combinations thereof: models A-H_2_O, A-I_2_, A-I_2_-H_2_O, A-I_3_^-^).

In model A-H_2_O, although geometry optimizations were initiated with the water molecules aligned at the center of the internal cavity (in the same manner that the iodine would be), the final result was that the water adhered to the inside walls of the helix, cf. Figure 1. The presence of such molecules inside the helix has not been commented on or hypothesized experimentally, even though hydration of amylose in principle is known to entail structural changes [51].

The binding of I_2_ to the helix (model A-I_2_ in Appendix A and Figure 1) leads to negligible distortions of the helix. The binding of interspersed water and iodine molecules (model A-I_2_-H_2_O) results in the water molecules migrating to the internal walls of the helix, suggesting that an iodine-water mixed chain is unlikely to occur in amylose. Indeed, I_2_ has been shown to efficiently bind to amylose in a dry/solid state, but much less so in solution (unless special conditions, e.g., heating, are provided) [31].

Attempts to optimize an A-I_3_^-^ model with several triiodide units inside the helix were unsuccessful, as all but one of the triiodide units migrated outside the helix upon geometry optimization. This may be taken as evidence against the hypothesis that the amylose-iodine/iodide blue color would be due to chains of I_3_^-^ units (and possibly, by extension, of I_n_^-^ units) aligned inside the helix. This finding is in line with the relatively hydrophobic nature of the helix cavity and with the anionic character of the triiodide. As a consequence, the A-I_3_^-^ model in Appendix A and Figure 1 only features a single I_3_^-^ unit inside the helix.

### 2.2. UV-Vis Simulations of Linear I_2_ Chains

Simulation of the UV-vis spectrum would allow one to verify which of the possible iodine/iodide combinations are the most likely to yield the ~600 nm maximum. This can be reliably done with TD-DFT calculations; however, since the amylose-iodine models were too large for such methods, Table 1 and Table 2 report TD-DFT calculations on linear iodine chains without, including the surrounding saccharide.

As shown in Table 1, an isolated iodine molecule displays a weak maximum at ~600 nm, due to a π* (HOMO) -> σ* (LUMO) transition. In a linear I_2_ dimer with I-I bonds identical to those in the monomer and with a set I_2_-I_2_ distance equal to the sum of the van der Waals radii, this maximum sees a bathochromic shift and a slight increase in intensity, as expected since at such short intermolecular distance the frontier orbitals of the two I_2_ molecules mix. Geometry optimization of this dimer leads to dissociation, and hence the ~600 nm peak is seen almost exactly in the same position as for the monomer—though, interestingly, the intensity of the “blue” band is twice larger than the monomer.

As also shown in Table 1, longer chains of I_2_ molecules follow the same trends seen when going from monomeric I_2_ to (I_2_)_2_ (including the elongation of intermolecular distances upon geometry optimization). The bathochromic and hyperchromic shifts continue to the extent that in the heptamer the intensity of the 600 nm band has ~ tripled compared with isolated iodine.

These data confirm the experimental observations and previous semiempirical calculations according to which a linear chain of I_2_ molecules may under certain conditions be hosted inside the amylose helix and that such a chain would display a more intense color than free iodine molecules in solution. The band responsible for this enhanced blue color entails a HOMO -> LUMO transition, where HOMO is a combination of I_2_ π* orbitals and LUMO is a combination of I_2_ σ* orbitals. The longer the poly-I_2_ chain is, the more there is a tendency for these molecular orbitals to no longer have exactly the same distribution across the same atoms–hence allowing the 600 nm band to gain a charge-transfer character which may be responsible for the increased oscillator strength. In the optimized poly-I_2_ geometries, the frontier molecular orbitals responsible for the blue color are located mostly at the two ends of the chains, and it is at these ends, and at the I_2_ molecules preceding the ends, that the above-mentioned charge transfer will occur. However, if the I_2_ molecules are compacted together within van der Waals radii (and not further optimized), it is the center and not the edges of the poly-I_2_ chain that holds the most contribution to the 600 nm band—though, again, the more important charge-transfer part will impact more clearly the iodine molecules preceding the ends of the chain.

### 2.3. UV-Vis Simulations of Linear I_n_^-^ Systems

Although the above considerations may explain the amylose–I_2_ interactions under certain conditions, the enhancement of the 600 nm band upon elongation of the poly-I_2_ is still relatively small—not even one order of magnitude upon going from monomer to heptamer. Therefore, poly-iodine-iodide chains were considered next, as illustrated in Table 2.

As seen in Table 2, the mono-anions I_n_^-^ (*n* = 1, 5, 7, 9) tend to exhibit bands in the ~400 nm region. However, the nature and intensity of this band depend strongly on the geometry of the anion. When the anion is built with equivalent I-I distances, to mimic complete delocalization of the negative charge across the molecule, the predicted absorption band remains very weak and close to the 400 nm region. The nature of this transition remains very similar to the one seen in the poly-I_2_ chains of Table 2 (i.e., π* (HOMO) -> σ* (LUMO))—except, of course, that it is no longer in the 600 nm region. These results are in line with previous lower level (i.e., semiempirical) INDO configuration interaction (CI) simulations relying on similar models, which were interpreted as evidence that poly-I_2_ but not iodine-iodide mixtures are hosted inside the helix in the blue amylose-iodine complex. However, Table 2 now also shows that when the geometry of the I_n_^-^ chains is optimized, the symmetry of the molecule is lost as the I-I distances no longer remain degenerate; importantly, this asymmetry allows the HOMO-LUMO transition to gain significant charge-transfer character—so much so that its intensity increases by 3–4 orders of magnitude compared with the completely symmetrical models. These increases in intensity are also accompanied by small bathochromic shifts; however, these shifts appear confined below 500 nm and hence these polyanions of iodide do not appear to be reasonable candidates for the experimentally observed 600 nm species. Another notable change in these geometry-optimized I_n_^-^ species is that at *n* > 3 the HOMO orbital involved in the visible transition now also has σ* character.

### 2.4. UV-Vis Simulations of Linear I_n_^-^-I_2_ Systems

Table 3 also shows a set of models which yield excellent agreement with the experiment, insofar as featuring extremely intense bands at 600 nm. These models include any I_n_^-^ species surrounded by two neutral I_2_ molecules. In such models, the ~600 nm transition occurs from a σ* orbital of the central oligo-iodide anion to the σ * orbitals of the terminal I_2_ molecules. For the I_2_-I_3_^-^I_2_ system, the formal structure featuring the I_2_ molecules at van der Waals radii from the I_3_^-^ anion does allow a very strong band at 680 nm. However, upon geometry optimization, this structure collapses into the I_7_^-^ anion, already seen in Table 2 as an unlikely candidate for the blue-colored complex. No such collapse is seen for the I_2_-I_5_^-^I_2_ species, which is thus the first of those discussed in the present study that can be proposed as a strong candidate as a contributor to the amylose-iodine blue color. Moreover, the average iodine-iodine distances in the DFT-optimized geometry of I_2_-I_5_^-^I_2_ is 3.09 Å, remarkably close to the 3.1 Å distance measured experimentally for the iodine-iodide-amylose complex. As shown in Table 3, I_2_ combinations with longer I_n_^-^ species cannot be excluded.

The 600 nm band in Table 3 is predicted to shift by as much as 25 nm upon going from *n* = 3 to *n* = 9, while over the same interval its intensity is seen to double. Given the uncertainties regarding the exact formula of the species, reliable extinction coefficients for the amylose-iodine-iodide complex are not known; Table 3 now shows that they would be particularly difficult to measure, since poly-iodine chains of various lengths will have different extinction coefficients. On the other hand, having singled out the I_2_-I_5_^-^I_2_ unit as the most likely candidate of those examined in Table 1, Table 2 and Table 3 for the “blue complex”, one may consider how long chains of such units may behave. Table 3 therefore also shows that the homologous assembly I_2_-I_5_^-^I_2_-I_5_^-^I_2_—technically, a dimer of I_2_-I_5_^-^I_2_, features an absorption maximum essentially at the same wavelength as the monomer—albeit with a different extinction coefficient. As such, I_2_-I_5_^-^I_2_ itself, and no longer chains/polymers thereof, appears sufficient to justify the blue color of the iodine-iodide-amylose complex.

### 2.5. Solvent Effects

The above considerations on TD-DFT spectra are based on vacuum calculations (ε = 1). The magnitude of solvent effects on the positions of UV-vis maxima are generally within the same margins as the effects of changing functionals and basis sets [58,59]. These margins are typically in the range of at most tens of nm. This is confirmed, as seen in Appendix A, by calculations on linear I_2_ and I_n_^-^ models. However, charge-transfer complexes may be distinctly more problematic to predict by DFT methods and can be distinctly more sensitive to the dielectric constants [59]. The linear I_2_ chains in Table 1 do show some charge transfer character as discussed above; hence, perhaps not surprisingly, Appendix A shows that, while the solvent effect is ~ 20 nm for the isolated I_2_ molecule, it almost doubles for the (I_2_)_2_ dimer. The strong solvent effect on the UV-vis spectrum of iodine is indeed well-known [60]. However, as seen in Appendix A for the representative case of I_2_-I_3_^-^I_2_ where charge transfer is even more dominant than in the neutral I_2_ homodimer, there is a very large ~exponential dependence on the position of the maximum (but not on the oscillator strength), so that between ε = 1 and ε = 9 a ~100 nm hypsochromic shift is observed—after which the changes are much smaller (i.e., only 25 nm from ε = 9 to ε~30, and only 4 nm further from ε ~ 30 to ε = 80.

The interior cavity/channel of the amylose helix has, as discussed above, been described as hydrophobic. Indeed, as also illustrated in Figure 1 and in Appendix A, this channel is lined with carbon-bound hydrogen and with ether oxygen atoms. Such an environment is somewhat reminiscent of the interior of proteins, but it is also similar to that provided by solvents such as diethyl ether, acetone, or dymethylsulfoxide; these solvents are in a range of ε ~ 4–30–which, according to our TD-DFT calculations, should be able to measurably affect the positions of the maxima. As discussed in the Introduction, subtle structural changes induced by other solutes, or by the origin of the amylose and hence the length of its chains, are known experimentally–and are also known to affect the UV-vis properties of the respective iodine/iodide complexes. It is reasonable to assume that at least part of these experimentally observed differences is due to slight changes in the dielectric constant within the internal amylose cavity as a consequence of slight changes to solvent exposure either at the ends of the chain (i.e., dependence on the length of the chain) or throughout the chain (i.e., from interactions with other solutes).

The strong solvent effects described here for iodine-iodide charge transfer complexes do mirror previous observations on other I_2_ charge transfer complexes [61].

## 3. Materials and Methods

UV–vis spectra were performed on Lambda 25 (PerkinElmer Singapore) spectrophotometers, in the range of 200–1000 nm, in 1 mL quartz cuvettes. Sample preparation and data collection take no more than two hours. Stock solutions included saturated starch, iodine aqueous solution 1 mM, iodine-iodide aqueous solution, iodide aqueous solution (100 mM), glucose (50 mM), 0.5% starch in water, 0.1 N ethanol, 10 mM methanol.

Models of the amylose helix were constructed based on a canonical structure extracted from https://www.biotopics.co.uk/jsmol/amylose-iodide.html (accessed on 2 August 2021), with the internal cavity left empty or filled with water, iodine and/or I_3_^-^. The guest molecules were placed and aligned along the central axis of the helix, at distances equal to the respective sums of the van der Waals radii. These models were subjected to geometry optimizations using molecular mechanics (SYBYL [62], MMFF [63]) and semiempirical (PM3, [64,65] PM6 [66] and AM1 [67]) methods as implemented within the Spartan software package [68]. The AM1 data are reported here; the other methods were found to yield qualitatively similar results. AM1 (Austin Model 1) is a semiempirical method based on modified neglect of differential diatomic overlap approximation, developed by Dewar and related to the PM3 method (Parametric Method 3) of Stewart and to its subsequent version PM6 [64,67]. We have previously shown that secondary-type structural elements in short chains of biopolymers (e.g., helical structures in peptides, or polylactic acid) are extremely challenging for all computational methods–even for density functional methods corrected/parametrized especially for describing weak/non-covalent interactions; nevertheless, even in those cases, semiempirical methods (and especially AM1 and PM6) performed close to on par with the highest-level density functional (DFT) methods [69,70,71,72]. Unfortunately, the size of the amylose-iodine models has so far precluded us from completing DFT calculations that would explicitly include the amylose helix. With these considerations in mind, the semiempirical (AM1) results are presented here only summarily, with the focus remaining on the density functional calculations on smaller, iodine-only models.

For iodine-only models (no helix included), geometries were built and, in cases indicated in Tables, optimized at the B3LYP/6-31G* level of theory in the Spartan software package. Further TD-DFT calculations were carried out using the B3LYP [73,74,75] functional coupled with def2-SV(P) [76] as implemented in the Gaussian software package; the PCM (polarizable continuum model) solvation model was employed with solvents as indicated in the text and Appendix A [77,78].

## 4. Conclusions

The amylose-iodine-iodide interaction has been modeled with electronic structure calculations. Changes in the structure of the helix upon iodine binding are predicted. Poly-I_2_ structures are shown to be responsible for the enhanced blue color under certain conditions (e.g., dry/solid amylose). Poly-I_n_^-^ structures are found unlikely to exist inside the amylose helix or to be responsible for the blue color. Instead, I_2_/I_n_^-^ pairs with charge transfer bands from the occupied I_n_^-^ (*n* > 3) σ* to the empty I_2_ σ * orbital are found to be reasonably responsible for the blue color. Of these, I_2_-I_5_^-^I_2_ associations are the smallest (and possibly–but not necessarily–repetitive) unit that represent local minima in DFT calculations, with average iodine-iodine distances essentially identical to the 3.1 Å value seen experimentally in the iodine-amylose complex. The distinct charge-transfer character of the UV-vis bands also brings about a strong dependence on the dielectric constant in the region ε ~ 1–30, which in turn may explain at least part of the dependence of the UV-vis properties of amylose-iodine/iodine complexes on various external factors that may subtly affect amylose architecture and hence exposure of the interior cavity to solvent (e.g., temperature, other solutes, solvents, chain length).

## Figures and Tables

**Figure 1 molecules-27-08974-f001:**
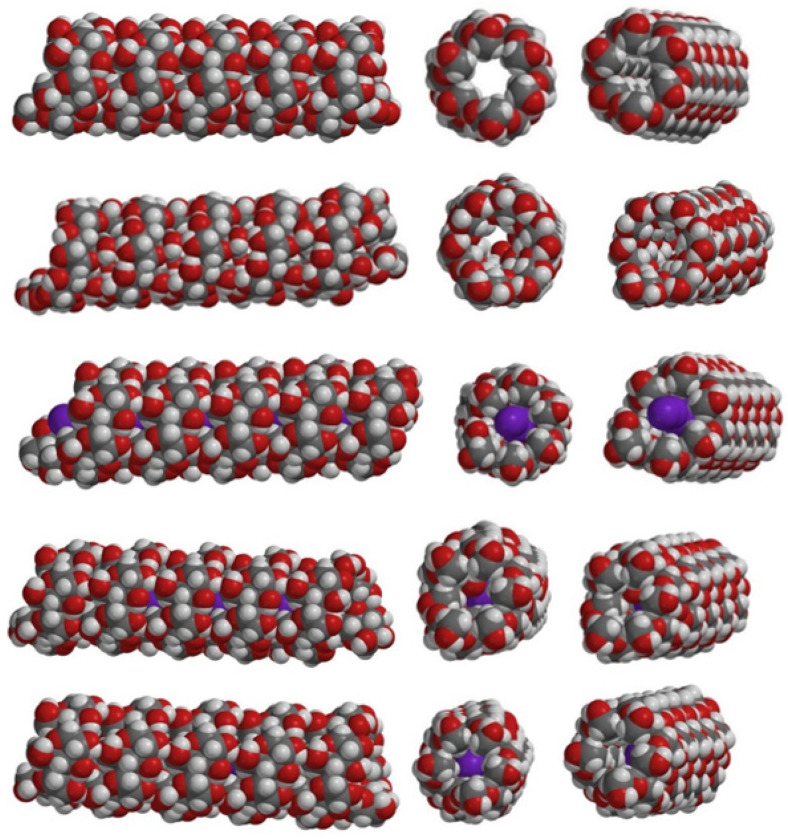
Structures of AM1-optimized amylose models, in order of rows: A, A-H_2_O, A-I_2_, A-I_2_-H_2_O, A-I_3_^-^.

**Table 1 molecules-27-08974-t001:** I-I distances (Å), wavelengths (λ, nm), oscillator strengths (OS), and relevant molecular orbitals for the largest visible maxima of linear I_2_ models (TD-DFT, B3LYP/def2-SV(P)). Data in italics are from energy calculations on structures with 2.72 Å I-I distances (known experimentally for I_2_ in solid state, identical to the those obtained upon DFT optimization of a single I_2_ molecule) [57], and with I_2_-I_2_ distances arbitrarily set at the limit of the sum of van der Waals radii (4.05 Å). All other data are from optimized geometries.

Model	I-I	λ	OS	Orbitals
I_2_	2.72	608	0.0014	HOMO (25) 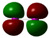 HOMO-1 (24) 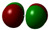	LUMO (26) 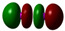
*(I_2_)_2_*	*2.72, 4.05*	*666*	*0.0016*	*HOMO-2 (48)* 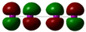 *HOMO-3 (47)* 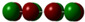	*LUMO (51)* 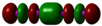
(I_2_)_2_	2.72, 12.97	607	0.0027	HOMO-2 (48) 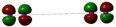 HOMO-3 (47) 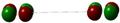	LUMO (51) 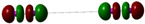
*(I_2_)_3_*	*2.72, 4.05*	*692*	*0.0019*	*HOMO-5 (70)* 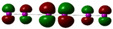	*LUMO (76)* 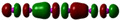
(I_2_)_3_	2.72, 7.62	609	0.0035	HOMO-2 (73) 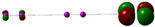 HOMO-3 (72) 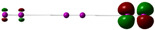	LUMO (77) 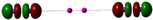
*(I_2_)_4_*	*2.72, 4.05*	*703*	*0.0022*	*HOMO-7 (93)* 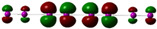	*LUMO (101)* 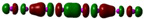
(I_2_)_4_	2.72, 6.06–6.11	612	0.0035	HOMO-2 (98) 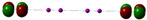 HOMO-3 (97) 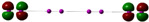	LUMO+2 (103) 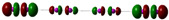
*(I_2_)_5_*	*2.72, 4.05*	*708*	*0.0026*	*HOMO-9 (116)* 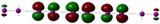	*LUMO (126)* 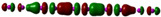
(I_2_)_5_	2.72, 6.48–6.56	610	0.0028	HOMO-2 (123) 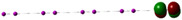 HOMO-3 (122) 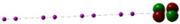	LUMO+3 (129) 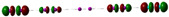
*(I_2_)_6_*	*2.72, 4.05*	*710*	*0.0028*	*HOMO-11 (139)* 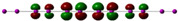	*LUMO (151)* 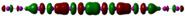
*(I_2_)_7_*	*2.72, 4.05*	*711*	*0.0030*	*HOMO-12 (163)* 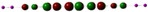 *HOMO-13 (162)* 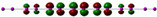	*LUMO (176)* 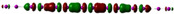
(I_2_)_7_	2.72, 6.20–6.29	611	0.0045	HOMO-1 (174) 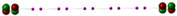 HOMO-3 (172) 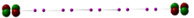	LUMO+5 (181) 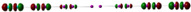

**Table 2 molecules-27-08974-t002:** I-I distances (Å), wavelengths for the major maximum in the visible region (nm), oscillator strengths (OS) and relevant molecular orbitals on linear I_n_^-^ models from TD-DFT (B3LYP/def2-SV(P)) calculations. Data shown in italics are from single-point calculations on structures built with intramolecular I-I distances known experimentally for I_2_ in solid state (2.72 Å). All other data are from optimized geometries.

Model	Distance	WaveLength	OS	Orbitals
*I_3_^-^*	*2.72*	*372*	*0.0000*	*HOMO (38)* * 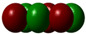 * *HOMO-1 (37)* * 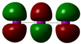 *	*LUMO (39)* * 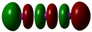 *
I_3_^-^	3.03	439	0.0018	HOMO-3 (35) 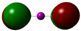 HOMO-4 (34) 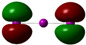	LUMO (39) 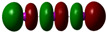
*I_5_^-^*	*2.72*	*365*	*0.0007*	*HOMO-6 (58)* * 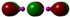 * *HOMO-7 (57)* * 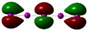 *	*LUMO (64)* * 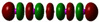 *
I_5_^-^	2.88, 3.19, 3.19, 2.88	387	2.9128	HOMO-4 (59) 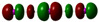	LUMO (64) 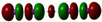
*I_7_^-^*	*2.72*	*391*	*5.1665*	*HOMO-5 (84)* * 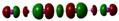 *	*LUMO (89)* * 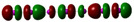 *
I_7_^-^	2.80, 3.37, 3.00, 3.00, 3.36, 2.80	479	3.7345	HOMO-5 (84) 	LUMO (89) 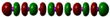
*I_9_^-^*	*2.72*	*458*	*3.6835*	*HOMO (114)* * 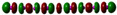 * *HOMO-2 (112)* * 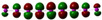 * *HOMO-5 (109)* * 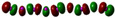 *	*LUMO (115)* * 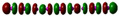 *
I_9_^-^	2.83, 3.29, 2.93, 3.07, 3.07, 2.93, 3.28, 2.83	431	0.4008	HOMO-3 (111) 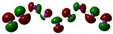	LUMO (115) 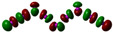

**Table 3 molecules-27-08974-t003:** I-I distances (Å), wavelengths for the major maximum in the visible region (nm), oscillator strengths (OS) and relevant molecular orbitals on linear I_n_^-^I_2_ models from TD-DFT (B3LYP/def2-SV(P)) calculations. Data shown in italics are for structures built with intramolecular I-I distances known experimentally for I_2_ in solid state (2.72 Å), and with intermolecular distances arbitrarily set at 4.05 Å (i.e., at the limit of the sum of van der Waals radii)–without any further geometry optimization. All other data are from fully optimized geometries.

Model	Distance	WaveLength	OS	Orbitals
*I_2_-I_3_^-^I_2_*	*2.72, 4.05*	*680*	*1.5507*	*HOMO-2 (86)* 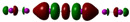	*LUMO (89)* 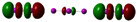
I_2_-I_3_^-^I_2_	2.80, 3.36, 3.00, 3.00, 3.36, 2.80	479	3.7407	HOMO-4 (84) 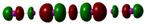	LUMO (89) 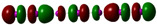
*I_2_-I_5_^-^I_2_*	*2.72, 4.05*	*681*	*1.9297*	*HOMO-4 (109)* 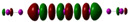	*LUMO (114)* 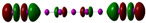
I_2_-I_5_^-^I_2_	2.76, 3.57, 2.89, 3.13, 3.13, 2.89, 3.57, 2.76	576	3.8021	HOMO-4 (109) 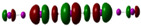	LUMO (114) 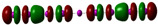
*I_2_-I_7_^-^I_2_*	*2.72, 4.05*	*666*	*2.6088*	*HOMO-4 (134)* 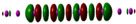	*LUMO (139)* 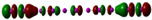
I_2_-I_9_^-^I_2_	2.72, 4.05	655	3.9582	HOMO-6 (157) 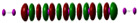	LUMO (164) 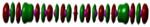
*I_2_-I_5_^-^I_2_-I_5_^-^I_2_*	*2.72, 4.05*	*681*	*2.3723*	*HOMO-9 (192)* 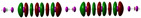	*LUMO+1 (203)* 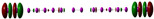

## Data Availability

Not applicable.

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
