# Peer review of "On the Origin of the Blue Color in The Iodine/Iodide/Starch Supramolecular Complex"

_molecules, 2022, doi:10.3390/molecules27248974_

Round 1

Reviewer 1 Report

This work has two parts, the first part is the structural and energetic analysis of the interaction between Amylose-iodine/iodide (sec 2.1). The second part is the UV-vis simulations. 

The first part is very questionable, since the semi-empirical methodology used usually does not correctly describe non-covalent interactions such as dispersion present in the complex. Was any correction used? Otherwise the results are doubtful. Is there evidence that semi-empirical methods can be used in this type of complex? These points should be taken into account by the authors.

Despite the approximation of the models, I consider that the spectroscopic analysis sheds light on the treated problem and even incorporates solvent effects.

In short, the authors should better justify the first part, or delete it.

Author Response

To some extent we might agree with the Reviewer. Therefore, the semiempirical part has now been drastically reduced, with some of it (Table 1) moved to Supporting Information. Additionally, some more technical details that place the performance of AM1 and PM3/PM6 in context have been added to the Methods section (including references to our previous assessments on the comparative performance of various computational methods, including semiempirical, on biopolymer secondary structure elements).

Reviewer 2 Report

The work entitled “On The Origin of the Blue Color in The Iodine / Iodide / Starch 2

Supramolecular Complex” by  S. Pesek et al. is an interesting piece of computational investigation on a rather iconic problem of general chemistry. The conclusions are interesting, as challenge the current belief, based on qualitative ideas.

Practically, the work can be published as it is, but I suggest to authors to be more explicit in certain details, going then to a minor revision.  For instance, by defining acronyms like INDO, AM1, PM3, PCM, etc. Please explain also why switched gaussian bases and codes between methods. For a supramolecular ensemble, one may expect to use long-range corrections (e.g. Grimme). This is not expected to change sensibly the spectral calculation, but may change the formation energy. Adding formation energies (Iodine polymers with respect to I2 or corresponding sum of I2 and I-) can be useful, for completeness, although not essential.

It may be useful to convert Tables 2 and 3 in figures, having then the graphic snippets magnified and annotating numbers around them, to gain better visibility for depicted orbitals.

Since semiempirical models like INDO, AM1, PM3 are no longer in frequent use, please briefly explain their specifics.  

Thus, I recommend publication after minor revision.

Author Response

  1. defining acronyms like INDO, AM1, PM3, PCM, etc. Since semiempirical models like INDO, AM1, PM3 are no longer in frequent use, please briefly explain their specifics. Please explain also why switched gaussian bases and codes between methods.

Reply: The manuscript has been revisited in order to make sure that acronyms are defined. As per Reviewer 1’s suggestion, the semiempirical part has now been drastically reduced. Nevertheless, some more technical details that place the performance of AM1 and PM3/PM6 in context have been added to the Methods section – including a more explicit statement on why the semiempirical methods were used for the very large helix+iodine models, while DFT was used for the smaller iodine-only models. In terms of switching between basis sets and codes for the DFT part: the Spartan implementation of B3LYP provided more straightforward convergence for geometry optimization than the Gaussian implementation; by contrast, the Gaussian interpretation either failed to reach convergence or failed to provide reasonable geometries even for the simplest of the models (e.g., I2). However, we found Spartan not versatile enough for UV-vis simulations – and hence ran the (single-point) TD-DFT simulations in Gaussian – using a compatible split valence with polarization functions on heavy atoms (since 6-31G itself is not available for iodine in Gaussian).

  1. For a supramolecular ensemble, one may expect to use long-range corrections (e.g. Grimme). This is not expected to change sensibly the spectral calculation, but may change the formation energy. Adding formation energies (Iodine polymers with respect to I2 or corresponding sum of I2 and I-) can be useful, for completeness, although not essential.

Reply: We have not added the formation energies, since upon close inspection of the optimized geometries, the intermolecular distances were already unsatisfactory (e.g., 12 A in the I2---I2 system). We have also tested Grimme’s corrections, but did not find any meaningful improvement in geometrical parameters.

  1. It may be useful to convert Tables 2 and 3 in figures, having then the graphic snippets magnified and annotating numbers around them, to gain better visibility for depicted orbitals.

Reply: It would be tempting to show larger versions of the molecular orbital plots. We would prefer to maintain the Table format for two reasons: (1) the orbitals do show repetitive features which may justify the condensed views in the Tables, and (2) with well over 50 orbitals to show, the resulting Figures may become too large and unmanageable themselves.

Round 2

Reviewer 1 Report

The revision improves this article. It must be accepted as such.